# Fast Axiomatic Attribution for Neural Networks

**Robin Hesse[1]**         **Simone Schaub-Meyer[1]**         **Stefan Roth[1,2]**

[1]Department of Computer Science, TU Darmstadt     [2]hessian.AI
{robin.hesse, simone.schaub, stefan.roth}@visinf.tu-darmstadt.de

## Abstract

Mitigating the dependence on spurious correlations present in the training dataset is a quickly emerging and important topic of deep learning. Recent approaches include priors on the feature attribution of a deep neural network (DNN) into the training process to reduce the dependence on unwanted features. However, until now one needed to trade off high-quality attributions, satisfying desirable axioms, against the time required to compute them. This in turn either led to long training times or ineffective attribution priors. In this work, we break this trade-off by considering a special class of *efficiently axiomatically attributable DNNs* for which an axiomatic feature attribution can be computed with only a single forward/backward pass. We formally prove that *nonnegatively homogeneous DNNs*, here termed $\mathcal{X}$-*DNNs*, are efficiently axiomatically attributable and show that they can be effortlessly constructed from a wide range of regular DNNs by simply removing the bias term of each layer. Various experiments demonstrate the advantages of $\mathcal{X}$-DNNs, beating state-of-the-art generic attribution methods on regular DNNs for training with attribution priors.

## 1   Introduction

Many traditional machine learning (ML) approaches, such as linear models or decision trees, are inherently explainable [4]. Therefore, an ML practitioner can comprehend why a method yields a particular prediction and correct the method if the explanation for the result is flawed. The prevailing ML architectures in use today [12, 25], namely deep neural networks (DNNs), unfortunately, do not come with this inherent explainability. This can cause models to depend on dataset biases and spurious correlations in the training data. For real-world applications, *e.g.*, credit score or insurance risk assignment, this can be highly problematic and potentially lead to models discriminating against certain demographic groups [3, 20]. To mitigate the dependence on spurious correlations in DNNs, attribution priors have been recently proposed [7, 22, 23]. By enforcing priors on the feature attribution of a DNN at training time, they allow actively controlling its behavior. As it turns out, attribution priors are a very flexible tool, allowing even complex model interventions such as making an object recognition model focus on shape [22] or less sensitive to high-frequency noise [7]. However, their use brings new challenges over regular training. First, computing the feature attribution of a DNN is a nontrivial task. It is critical to use an attribution method that faithfully reflects the true behavior of the deep network and ideally satisfies the axioms proposed by Sundararajan et al. [34]. Otherwise, spurious correlations may go undetected and the attribution prior would be ineffective. Second, since the feature attribution is used in each training step, it needs to be efficiently computable. Existing work incurs a trade-off between high-quality attributions for which formal axioms hold [34] and the time required to compute them [7]. Prior work on attribution priors thus had to choose whether to rely on high-quality feature attributions *or* allow for efficient training. In this work, we obviate this trade-off.

---

Code and additional resources at https://visinf.github.io/fast-axiomatic-attribution/.

Specifically, we make the following contributions: *(i)* We propose to consider a special class of DNNs, termed *efficiently axiomatically attributable DNNs*, for which we can compute a closed-form axiomatic feature attribution that satisfies the axioms of Sundararajan et al. [34], requiring only one gradient evaluation. As a result, we can compute axiomatic high-quality attributions with only one forward/backward pass, and hence, require only a fraction of the computing power that would be needed for regular DNNs, which involve a costly numerical approximation of an integral [7, 34]. This significant improvement in efficiency makes our considered class of DNNs particularly well suited for scenarios where the attribution is used in the training process, such as for training with attribution priors. *(ii)* We formally prove that *nonnegatively homogeneous DNNs* (termed $\mathcal{X}$-DNNs) are efficiently axiomatically attributable DNNs and establish a new theoretical connection between Input×Gradient [27] and Integrated Gradients [34] for nonnegatively homogeneous DNNs of different degrees. *(iii)* We show how $\mathcal{X}$-DNNs can be instantiated from a wide range of regular DNNs by simply removing the bias term of each layer. While this may seem like a significant restriction, we show that the impact on the predictive accuracy in two different application domains is surprisingly minor. In a variety of experiments, we demonstrate the advantages of $\mathcal{X}$-DNNs, showing that they *(iv)* admit accurate axiomatic feature attributions at a fraction of the computational cost and *(v)* beat state-of-the-art generic attribution methods for training regular networks with an attribution prior.

## 2 Related work

**Attribution methods** can roughly be divided into perturbation-based [13, 39, 41, 42] and backpropagation-based [1, 5, 28, 30, 34] methods. The former repeatably perturb individual inputs or neurons to measure their impact on the final prediction. Since each perturbation requires a separate forward pass through the DNN, those methods can be computationally inefficient [28] and consequently inappropriate for inclusion into the training process. We thus consider *backpropagation-based methods* or, more precisely, gradient-based and rule-based attribution methods. They propagate an importance signal from the DNN output to its input using either the gradient or predefined rules, making them particularly efficient [28], and thus, well suited for inclusion into the training process. Gradient-based methods have the advantage of scaling to high-dimensional inputs, can be efficiently implemented using GPUs, and directly applied to any differentiable model without changing it [2].

The saliency method [30], defined as the absolute input gradient, is an early gradient-based attribution method for DNNs. Shrikumar et al. [27] propose the Input×Gradient method, *i.e.*, weighting the (signed) input gradient with the input features, to improve sharpness of the attributions for images. Bach et al. [5] introduce the rule-based Layerwise Relevance Propagation (LRP), with predefined backpropagation rules for each neural network component. As it turns out, LRP without modifications to deal with numerical instability can be reduced to Input×Gradient for DNNs with ReLU [19] activation functions [1, 27], and hence, can be expressed in terms of gradients as well. DeepLIFT [28] is another rule-based approach similar to LRP, relying on a neutral baseline input to assign contribution scores relative to the difference of the normal activation and reference activation of each neuron. Generally, rule-based approaches have the disadvantage that each DNN component requires custom modules that may have no GPU support and require a model re-implementation.

**Axiomatic attributions.** As it is hard to empirically evaluate the quality of feature attributions, Sundararajan et al. [34] propose several axioms that high-quality attribution methods should satisfy:

**Sensitivity (a)** is satisfied if for every input and baseline that differ in one feature but have different predictions, the differing feature should be given a non-zero attribution.

**Sensitivity (b)** is satisfied if the function implemented by the deep network does not depend (mathematically) on some variable, then the attribution to that variable is always zero.

**Implementation invariance** is satisfied if the attributions for two functionally equivalent networks are always identical.

**Completeness** is satisfied if the attributions add up to the difference between the output of the network for the input and for the baseline.

**Linearity** is satisfied if the attribution of a linearly composed deep network $aF_1 + bF_2$ is equal to the weighted sum of the attributions for $F_1$ and $F_2$ with weights $a$ and $b$, respectively.

**Symmetry preservation** is satisfied if for all inputs and baselines that have identical values for symmetric variables, the symmetric variables receive identical attributions.

[34] shows that none of the above methods satisfy all axioms, *e.g.*, the saliency method and Input×Gradient can suffer from the well-known problem of gradient saturation, which means that even important features can have zero attribution. To overcome this, [34] introduces Integrated Gradients, a gradient-based backpropagation method that provably satisfies these axioms; it is considered a high-quality attribution method to date [7, 14]. Its crucial disadvantage over previous methods is that an integral has to be solved, which generally requires an approximation based on $\sim 20$–$300$ gradient calculations, making it correspondingly computationally more expensive than, *e.g.*, Input×Gradient.

**Attribution priors.** The above attribution methods cannot only be used for explaining a model's behavior but also to actively control the behavior. To that end, the training objective can be formulated as

$$\theta^* = \arg\min_\theta \frac{1}{|X|} \sum_{(x,y)\in X} \mathcal{L}(F_\theta; x, y) + \lambda\Omega(\mathcal{A}(F_\theta, x)), \tag{1}$$

where a model $F_\theta$ with parameters $\theta$ is trained on the annotated dataset $X$. $\mathcal{L}$ denotes the regular task loss, and $\Omega$ is a scalar-valued loss of the feature attribution $\mathcal{A}$, which is called the attribution prior [7]; $\lambda$ controls the relative weighting. For example, by forcing certain values of the attribution to be zero, we can mitigate the dependence on unwanted features [23]. But also more complex model interventions, such as making an object recognition model focus on shape [22] or less sensitive to high-frequency noise [7], can be formulated using attribution priors.

An early instance of this idea is the Right for the Right Reasons (RRR) approach of Ross et al. [23], which uses the input gradient of the *log* prediction to mitigate the dependence on unwanted features. While this is more stable than simply using the input gradient, it still suffers from the problem of saturation. RRR may thus not reflect the true behavior of the model, and therefore, miss relevant features. Subsequent work addresses this issue using axiomatic feature attribution methods, specifically Integrated Gradients [6, 14, 34], which allows for more effective attribution priors [7] but incurs significant computational overhead, rendering them impractical for many scenarios. For example, Liu and Avci [14] report a thirty-fold increase in training time compared to standard training. Rieger et al. [22] propose an alternative attribution prior based on a rule-based contextual decomposition [18, 31] (CD) as attribution method. This allows to consider clusters of features [7] instead of individual features and to define attribution priors working on feature groups. However, computing the attribution for individual features becomes computationally inefficient [7]. Additionally, since CD is a rule-based attribution method, it requires custom modules and cannot be applied to all types of DNNs [7]. The very recently proposed Expected Gradients [7] method reformulates Integrated Gradients as an expectation, allowing a sampling-based approximation of the attribution. Erion et al. [7] argue that similar to batch gradient descent, where the true gradient of the loss function is approximated over many training steps, the sampling-based approximation allows to approximate the attribution over many training steps. This results in better attributions while using fewer approximation steps. Even using as little as one reference sample, *i.e.*, only one gradient computation, can yield advantages over the regular input gradient. However, we show that using only one reference sample still does not yield the same attribution quality as an axiomatic feature attribution method, reflected in less effective attribution priors. Schramowski et al. [26] propose a human-in-the-loop strategy to define appropriate attribution priors while training. Our attribution method is complementary and could be used within their framework.

## 3 Efficiently axiomatically attributable DNNs

Formally, given a function $F\colon \mathbb{R}^n \mapsto \mathbb{R}$, representing a single output node of a DNN, and an input $x \in \mathbb{R}^n$, the feature attribution for the prediction at input $x$ relative to a baseline input $x'$ is a vector $\mathcal{A}(F, x, x') \in \mathbb{R}^n$, where each element $a_i$ is the contribution of feature $x_i$ to the prediction $F(x)$ [34]. In this work we seek an attribution method that is particularly well suited for scenarios where such an attribution is used at training time, *e.g.*, training with attribution priors. As such, the attribution method should be of high quality, while being efficiently computable in a single forward/backward pass. Since attributions obtained from Integrated Gradients [34] have strong theoretical justifications and are known to be of high-quality [14], they will serve as our starting point. In general, however, Integrated Gradients are expensive to compute for arbitrary DNNs. Therefore, in this work, we restrict our attention to a special sub-class of DNNs, termed *efficiently axiomatically attributable DNNs*, that require only a single forward/backward pass to compute a closed-form solution of Integrated Gradients. We show that nonnegatively homogeneous DNNs belong to this class and use this insight

to guide the design of a concrete instantiation of efficiently axiomatically attributable DNNs. While there may be several such instantiations, we chose this particular one as it can be easily constructed from a wide range of regular DNNs by simply removing the bias term of each layer. This ensures comparability to prior work and allows for an easy adaptation of existing network architectures.

**Definition 3.1.** We call a DNN $F\colon \mathbb{R}^n \mapsto \mathbb{R}$ *efficiently axiomatically attributable w.r.t.* a baseline $x' \in \mathbb{R}^n$, if there exists a closed-form solution of the axiomatic feature attribution method Integrated Gradients $\mathrm{IG}_i(F, x, x')$ along the $i^{\text{th}}$ dimension of $x \in \mathbb{R}^n$, requiring only one forward/backward pass.

Note that all differentiable models are efficiently axiomatically attributable *w.r.t.* the trivial baseline $x' = x$. However, using such a baseline is not helpful. Instead, commonly chosen baselines are some kind of averaged input features or baselines such that $F(x') = 0$, which allow an interpretation of the attribution that amounts to distributing the output to the individual input features [34].

**Proposition 3.2.** *For a DNN $F\colon \mathbb{R}^n \mapsto \mathbb{R}$ there exists a closed-form solution of $\mathrm{IG}_i(F, x, \mathbf{0})$* w.r.t. *the zero baseline $\mathbf{0} \in \mathbb{R}^n$, requiring only one forward/backward pass, if $F$ is strictly positive homogeneous of degree $k \in \mathbb{R}_{\geq 1}$*, i.e., *$F(\alpha x) = \alpha^k F(x)$ for $\alpha \in \mathbb{R}_{>0}$.*

*Proof.* Sundararajan et al. [34] define the axiomatic feature attribution method Integrated Gradients (IG) along the $i^{\text{th}}$ dimension for a given model $F$, input $x$, baseline $\mathbf{0}$, and straightline path $\gamma(\alpha) = \alpha x$ as

$$\mathrm{IG}_i(F, x, \mathbf{0}) = \int_0^1 \frac{\partial F(\gamma(\alpha))}{\partial \gamma_i(\alpha)} \frac{\partial \gamma_i(\alpha)}{\partial \alpha} \, d\alpha = \int_0^1 \frac{\partial F(\alpha x)}{\partial \alpha x_i} \frac{\partial \alpha x_i}{\partial \alpha} \, d\alpha \,. \tag{2}$$

Assuming $F$ is strictly positive homogeneous of degree $k \geq 1$, we can write Integrated Gradients in Eq. (2) as

$$\mathrm{IG}_i(F, x, \mathbf{0}) = \lim_{\beta \to 0} \int_\beta^1 \frac{\partial F(\alpha x)}{\partial \alpha x_i} x_i \, d\alpha = \lim_{\beta \to 0} \int_\beta^1 \alpha^{k-1} \frac{\partial F(x)}{\partial x_i} x_i \, d\alpha = \frac{1}{k} x_i \frac{\partial F(x)}{\partial x_i} \,. \tag{3}$$

The gradient expression in Eq. (3) can be computed using a single forward/backward pass. $\qquad\square$

While Ancona et al. [1] already found that Input×Gradient equals Integrated Gradients with the zero baseline for linear models or models that behave linearly for a selected task, our Proposition 3.2 is more general: We only require strictly positive homogeneity of an arbitrary order $k \geq 1$. This allows us to consider a larger class of models including nonnegatively homogeneous DNNs, which generally are not linear.

**Definition 3.3.** We call a DNN $F\colon \mathbb{R}^n \mapsto \mathbb{R}$ *nonnegatively homogeneous*, if $F(\alpha x) = \alpha F(x)$ for all $\alpha \in \mathbb{R}_{\geq 0}$.

**Corollary 3.4.** *Any nonnegatively homogeneous DNN is efficiently axiomatically attributable* w.r.t. *the zero baseline $\mathbf{0} \in \mathbb{R}^n$ and a closed-form solution of the axiomatic feature attribution method Integrated Gradients, requiring only one forward/backward pass, exists.*

*Proof.* Corollary 3.4 follows directly from Proposition 3.2 and Definitions 3.1 and 3.3. $\qquad\square$

**Definition 3.5.** We let $\mathcal{X}$-*DNN* denote a nonnegatively homogeneous DNN. Further, for any $\mathcal{X}$-DNN $F\colon \mathbb{R}^n \mapsto \mathbb{R}$, we let $\mathcal{X}$-*Gradient* ($\mathcal{X}$G) be an axiomatic feature attribution method relative to the zero baseline $\mathbf{0} \in \mathbb{R}^n$ defined as

$$\mathcal{X}\mathrm{G}_i(F, x) = \mathrm{IG}_i(F, x, \mathbf{0}) = x_i \frac{\partial F(x)}{\partial x_i} \,. \tag{4}$$

Note that while the formulas for the existing attribution method Input×Gradient [27] and our novel $\mathcal{X}$-Gradient are equal, $\mathcal{X}$-Gradient is only defined for the subclass of $\mathcal{X}$-DNNs and provably satisfies axioms that are generally not satisfied by Input×Gradient. Additionally, from the nonnegative homogeneity of $\mathcal{X}$-DNNs it follows that $\mathcal{X}$-Gradient attributions are also nonnegatively homogeneous. This allows us to define another desirable axiom that is in line with intuition about how attribution should work and that is satisfied by $\mathcal{X}$-Gradient.

**Definition 3.6.** An attribution method $\mathcal{A}$ satisfies *nonnegative homogeneity* if $\mathcal{A}(F, \alpha x, \alpha x') = \alpha \, \mathcal{A}(F, x, x')$ for all $\alpha \in \mathbb{R}_{\geq 0}$.

Table 1: *Overview of different gradient-based DNN attribution methods and the axioms [34] that they provably satisfy.* The left-hand side methods (Integrated Gradients, Expected Gradients) induce one to two orders of magnitude of computational overhead compared to the methods on the right-hand side. The methods on the right-hand side require only one gradient evaluation (indicated by (1) for Expected Gradients with one reference sample), and thus, can be computed in a single forward/backward pass. Note how $\mathcal{X}$-Gradient satisfies all axioms while requiring as little computational cost as a simple gradient evaluation, however being only defined for $\mathcal{X}$-DNNs.

| Axiom | Integrated Gradients | Expected Gradients | Expected Gradients(1) | (Input $\times$) Gradient | $\mathcal{X}$-Gradient |
|---|---|---|---|---|---|
| *Sensitivity (a)* | ✓ | ✓ | ✗ | ✗ | ✓ |
| *Sensitivity (b)* | ✓ | ✓ | ✓ | ✓ | ✓ |
| *Implementation invariance* | ✓ | ✓ | ✗ | ✓ | ✓ |
| *Completeness* | ✓ | ✓ | ✗ | ✗ | ✓ |
| *Linearity* | ✓ | ✓ | ✗ | ✓ | ✓ |
| *Symmetry-preserving* | ✓ | ✓ | ✗ | ✓ | ✓ |

For an overview of the axioms [34] that are satisfied by popular gradient-based attribution methods, see Table 1. The right-hand side methods use only one gradient evaluation, and therefore, have similar computational expense. The left-hand side methods generally require multiple gradient evaluations until convergence, making them correspondingly computationally more expensive. Note that $\mathcal{X}$-Gradient satisfies all the axioms satisfied by Integrated Gradients and Expected Gradients [7], assuming convergence of the latter, while requiring only a fraction of the computational cost, however being only defined for $\mathcal{X}$-DNNs. Existing methods that have similar computational expense as $\mathcal{X}$-Gradient generally do not satisfy all of the axioms, and therefore, are likely to produce lower quality attributions, which can be misleading and less effective for imposing attribution priors.

**Constructing $\mathcal{X}$-DNNs.** With this motivation in mind, we will now study concrete instantiations of nonnegatively homogeneous DNNs. Note that this class of DNNs has already been considered by Zhang et al. [40], however, neither at the same level of detail nor in the context of feature attributions. We define the output of a regular feedforward DNN $F\colon \mathbb{R}^n \mapsto \mathbb{R}^o$, for an input $x \in \mathbb{R}^n$, as a recursive sequence of layers $l$ that are applied to the output of the respective previous layer:

$$F_l\left(x\right) = \begin{cases} \psi_l\left(\phi_l\left(W_l F_{l-1}(x) + b_l\right)\right) & \text{if } l \geq 1 \\ x & \text{if } l = 0, \end{cases} \tag{5}$$

with $W_l$ and $b_l$ being the weight matrix and bias term for layer $l$, $\phi_l$ being the corresponding activation function, and $\psi_l$ being the corresponding pooling function. Both $\phi_l$ and $\psi_l$ are optional; alternatively they are the identity function. For simplicity, we assume that the last task-specific layer, *e.g.*, the softmax function for classification tasks, is part of the loss function. Further, for a cleaner notation that aligns with [34], we assume *w.l.o.g.* that we are only considering one output node at a time, *e.g.*, the logit of the target class for classification tasks. This yields the DNN $F\colon \mathbb{R}^n \mapsto \mathbb{R}$ that we consider and allows us to directly compute the derivative of the model *w.r.t.* an input feature $x_i$. Importantly, the above formalization comprises many popular layer types and architectures. For example, fully connected and convolutional layers are essentially matrix multiplications [36], and therefore, can be expressed by Eq. (5). Skip connections can also be expressed as matrix multiplication by appending the identity matrix to the weight matrix so that the input is propagated to later layers [36]. This allows us to describe even complex architectures such as the ResNet [9] variant proposed by [40]. As the above definition of a DNN includes models that are generally not nonnegatively homogeneous, we have to make some assumptions.

**Assumption 3.7.** *The activation functions $\phi_l$ and pooling functions $\psi_l$ in the model are nonnegatively homogeneous. Formally, for all $\alpha \in \mathbb{R}_{\geq 0}$:*

$$\alpha\phi_l(z) = \phi_l(\alpha z) \quad and \quad \alpha\psi_l(z) = \psi_l(\alpha z). \tag{6}$$

**Proposition 3.8.** *Piecewise linear activation functions with two intervals separated by zero satisfy Assumption 3.7. For $z = (z_1, \ldots, z_n) \in \mathbb{R}^n$, these activation functions $\phi_l\colon \mathbb{R}^n \mapsto \mathbb{R}^n$ are defined as*

$$\phi_l\left(z\right) = (\phi_l'(z_1), \ldots, \phi_l'(z_n)) \quad with \quad \phi_l'(z_i) = \begin{cases} a_{l,1} z_i & \text{if } z_i > 0 \\ a_{l,2} z_i & \text{if } z_i \leq 0. \end{cases} \tag{7}$$

**Proposition 3.9.** *Linear pooling functions or pooling functions selecting values based on their relative ordering satisfy Assumption 3.7. For $z = (z_1, \ldots, z_n) \in \mathbb{R}^n$, these pooling functions $\psi_l \colon \mathbb{R}^n \mapsto \mathbb{R}^m$ are defined as*

$$\psi_l(z) = (\psi'_l(z'_1), \ldots, \psi'_l(z'_m)), \tag{8}$$

*with $z'_i$ being a grouping of entries in $z$ based on their spatial location and $\psi'_l \colon \mathbb{R}^m \mapsto \mathbb{R}$ being linear or a selection of a value based on its relative ordering,* e.g.*, the maximum or minimum value.*

For proofs of Propositions 3.8 and 3.9, please refer to the supplemental material. Activation functions in Proposition 3.8 include ReLU [19], Leaky ReLU [16], and PReLU [8]. Linear pooling functions in Proposition 3.9 include average pooling, global average pooling, and strided convolutions. Other pooling functions in Proposition 3.9 include max pooling and min pooling [32], where the largest or smallest value is selected. Therefore, DNN architectures satisfying Assumption 3.7 include, inter alia, AlexNet [11], VGGNet [29], ResNet [9] as introduced in [40], and MLPs with ReLU activations. They alone have been cited well over one hundred thousand times, showing that we are considering a substantial fraction of commonly used DNN architectures. However, these architectures are generally still not nonnegatively homogeneous. It is easy to see that even for a simple linear model $F(x) = ax + b$ that can be expressed by Eq. (5) and that satisfies Assumption 3.7, nonnegative homogeneity does not hold, because $0F(x) = 0 \neq b = F(0x)$. Therefore, in a final step we set the bias term of each layer to zero. As this may seem like a significant restriction, we show in Sec. 4 that the impact on the predictive accuracy in two different application domains is surprisingly minor.

**Corollary 3.10.** *Any regular DNN given by Eq. (5) satisfying Assumption 3.7 can be transformed into an $\mathcal{X}$-DNN by removing the bias term of each layer.*

*Proof.* A DNN $F$ with $L$ layers given by Eq. (5) with all biases $b_l$ set to 0 can be written as $F(x) = \psi_L(\phi_L(W_L(\ldots(\psi_1(\phi_1(W_1 x))))))$. As all the pooling functions $\psi_l$, activation functions $\phi_l$, and matrix multiplications $W_l$ in $F$ are nonnegatively homogeneous, it follows that $F(\alpha x) = \alpha F(x)$ for all $\alpha \in \mathbb{R}_{\geq 0}$. $\qquad\square$

**Further discussion.** We additionally note that our results have interesting consequences for DNNs in certain application domains, *e.g.*, in computer vision, as they allow to relate efficient axiomatic attributability to desirable properties of DNNs:

**Remark 3.11.** If a DNN $F \colon \mathbb{R}^n \mapsto \mathbb{R}$, taking an image $x \in \mathbb{R}^n$ as input, is equivariant *w.r.t.* to the image contrast, it is efficiently axiomatically attributable.

This observation follows directly from the fact that contrast equivariance implies nonnegative homogeneity. Consequentially, contrast-equivariant DNNs for regression tasks, such as image restoration or image super-resolution, are automatically efficiently axiomatically attributable. For classification tasks, such as image classification or semantic segmentation, contrast equivariance of the logits at the output implies efficient axiomatic attributability. If the classification is done using a softmax, then this also implies contrast invariance of the classifier output. In other words, there is a close relation between efficient axiomatic attributability and the desirable property of contrast equi-/invariance. We further illustrate this experimentally in Sec. 4.4.

**Limitations.** So far, we have discussed the advantages of $\mathcal{X}$-DNNs such as being able to efficiently compute high-quality feature attributions. However, we also want to mention the limitations of our method. First, our method can only be applied to certain DNNs satisfying Assumption 3.7. Although this is a large class of models, our method is not completely model agnostic as other gradient-based attribution methods. Second, removing the bias term might be disadvantageous in certain scenarios. Intuitively, the bias term can be seen like an intercept in a linear function, and therefore, is important to fit given data. As a matter of fact, a DNN without bias terms will always produce a zero output for the zero input, which might be problematic. Additionally, prior work argues that the bias term is an important factor for the predictive performance of a DNN [36]. However, these theoretical foundations are somewhat contradictory to our own practical findings. In Sec. 4, we show that removing the bias term has less of a negative impact than perhaps expected, indicating that removing it can be a plausible intervention on the model architecture. As we cannot guarantee that this holds for all DNNs, we recommend that practitioners who plan on using our method, first make a preliminary analysis of whether removing the bias from the model at hand is plausible. Third, our method uses implicitly the zero baseline $\mathbf{0}$. As $F(\mathbf{0}) = 0$, this is a reasonable choice because it can be interpreted as being neutral [34]. Nevertheless, other baselines could produce attributions that are

Table 2: *Top-5 accuracy* on the ImageNet [24] validation split and mean absolute relative difference (see supplemental material) of Input×Gradient for regular DNNs resp. $\mathcal{X}$-Gradient for $\mathcal{X}$-DNNs to the numerical approximation of Integrated Gradients. Note how removing the bias ($\mathcal{X}$-DNN) impairs the accuracy only marginally while reducing the mean absolute relative difference to Integrated Gradients significantly, confirming our theoretical finding that $\mathcal{X}$-Gradient equals Integrated Gradients.

| Model | Top-5 accuracy (%, ↑) | | | Mean absolute relative difference (%, ↓) | | |
| | AlexNet | VGG16 | ResNet-50 | AlexNet | VGG16 | ResNet-50 |
| --- | --- | --- | --- | --- | --- | --- |
| Regular DNN | **79.21** | **90.44** | **92.56** | 79.0 | 97.8 | 93.8 |
| $\mathcal{X}$-DNN | 78.54 | 90.25 | 91.12 | **1.2** | **0.4** | **0.0** |

better suited for certain tasks [21, 33, 37]. For example, the zero baseline will generally assign lower attribution scores to features closer to zero, which can result in misleading attributions. Whether the advantages outweigh the disadvantages must be decided for each application, individually. In Sec. 4 we demonstrate the advantages of $\mathcal{X}$-DNNs, beating state-of-the-art generic attribution methods for training with attribution priors.

## 4 Experiments

To demonstrate the practicability of our proposed method, we now evaluate it in various experiments using two different data domains to confirm the following points: *(1)* It is plausible to remove the bias term in order to obtain $\mathcal{X}$-DNNs. *(2)* Our $\mathcal{X}$-Gradient method produces superior attributions compared to other efficient gradient-based attribution methods. *(3)* Our $\mathcal{X}$-Gradient method has advantages over state-of-the-art generic attribution methods for training with attribution priors. *(4)* $\mathcal{X}$-DNNs are robust to multiplicative contrast changes.

**Experimental setup.** For our experiments on models for image classification, *i.e.*, Section 4.1, 4.2 and 4.4, we use the ImageNet [24] dataset, containing about 1.2 million images of 1000 different categories. We train on the training split and report numbers for the validation split. In Sec. 4.2 we quantify the quality of attributions for image classification models by adapting the metrics proposed by Lundberg et al. [15] to work with image data. These metrics reflect how well an attribution method captures the relative importance of features by measuring the network's accuracy or its output logit of the target class while masking out a progressively increasing fraction of the features based on their relative importance. For example, for the Keep Positive Mask (KPM) metric, the output logit of the target class should stay as high as possible while progressively masking out the least important features. As a mask we use a Gaussian blur of the original image. For a detailed description of the metrics, please refer to [15] or the supplemental material. If not indicated otherwise, we assume numerical convergence for Integrated Gradients and Expected Gradients, which we found to occur after $\sim 128$ approximation steps (see supplemental material).

### 4.1 Removing the bias term in DNNs

Historically, the bias term plays an important role and almost all DNN architectures use one. In this first experiment, we evaluate how much removing the bias to obtain an $\mathcal{X}$-DNN affects the accuracy of different DNNs. To this end, we train multiple popular image classification networks, AlexNet [11], VGG16 [29], and the ResNet-50 variant of [40], as well as their corresponding $\mathcal{X}$-DNN variants obtained by removing the bias term, on the challenging ImageNet [24] dataset. The resulting top-5 accuracy on the validation split is given in Table 2. As we can observe, removing the bias decreases the accuracy of the models only marginally. This is a somewhat surprising result since prior work indicates that the bias term in DNNs plays an important role [36]. We hypothesize that when removing the bias term, the DNN learns some kind of layer averaging strategy that compensates for the missing bias. For an additional comparison between a DNN with bias and its corresponding $\mathcal{X}$-DNN in a non-vision domain, see Sec. 4.3, which mirrors our findings here. Additionally, to empirically validate our finding that $\mathcal{X}$-Gradient ($\mathcal{X}G$) equals Integrated Gradients for $\mathcal{X}$-DNNs, we report the mean absolute relative difference (see supplemental material) between the attribution obtained from Integrated Gradients [34] and the attribution obtained from computing Input×Gradient for regular DNNs resp. $\mathcal{X}$-Gradient for $\mathcal{X}$-DNNs over the ImageNet validation split. For regular models with

Table 3: *Metrics of Lundberg et al. [15] to measure the attribution quality* of different attribution methods. Please refer to the experimental setup in the beginning of Sec. 4 and the supplemental material for an introduction of the metrics. We evaluate Integrated Gradients (IG) [34], random attributions (Random), input gradient attributions (Grad), Expected Gradients (EG) [7], and our novel $\mathcal{X}$-Gradient ($\mathcal{X}$G) attribution on a regular AlexNet [40] and the corresponding $\mathcal{X}$-AlexNet. The numbers in parentheses indicate the required gradient calls. Our method is on par with IG in terms of quality while requiring two orders of magnitude less computational power.

| Method | AlexNet | | | | $\mathcal{X}$-AlexNet | | | |
| --- | --- | --- | --- | --- | --- | --- | --- | --- |
| | KPM ↑ | KNM ↓ | KAM ↑ | RAM ↓ | KPM ↑ | KNM ↓ | KAM ↑ | RAM ↓ |
| IG (128) | **7.57** | **1.67** | **25.22** | **11.12** | 7.38 | 2.21 | 21.79 | 11.68 |
| Random | 3.68 | 3.68 | 14.12 | 14.10 | 3.81 | 3.81 | 13.52 | 13.50 |
| Grad (1) | 3.62 | 3.88 | 20.78 | 11.82 | 3.87 | 4.34 | 19.75 | **11.25** |
| EG (1) | 4.92 | 2.97 | 20.49 | 13.76 | 5.41 | 3.19 | 19.47 | 13.19 |
| $\mathcal{X}$G (1) | N/A | N/A | N/A | N/A | **7.38** | **2.21** | **21.83** | 11.68 |

biases, Integrated Gradients produce a very different attribution compared to Input×Gradient. For $\mathcal{X}$-DNNs on the other hand, the two attribution methods are virtually identical, as expected. The small deviation can be explained by the fact that the result of Integrated Gradients [34] is computed via numerical approximation, whereas our method computes the exact integral (of course only for $\mathcal{X}$-DNNs). We make the pre-trained $\mathcal{X}$-DNN models publicly available to promote a wide adoption of efficiently axiomatically attributable models.

## 4.2 Benchmarking gradient-based attribution methods

As prior work [7, 14] but also our experiment in Sec. 4.3 suggest that the quality of an attribution method positively impacts the effectiveness of attribution priors, we benchmark our method against existing gradient-based attribution methods that are commonly used for training with attribution priors. For evaluation, we use the metrics from [15] adapted to work with image data. Using these metrics allows for a diverse assessment of the feature importance [15] and ensures consistency with the experimental setup in [7]. Table 3 shows the resulting numbers for a regular AlexNet and our corresponding $\mathcal{X}$-AlexNet. Due to the axioms satisfied by the Integrated Gradients method, it produces the best attributions for the regular network, which is in line with the results in [38]. However, as it approximates an integral where each approximation step requires an additional gradient evaluation, it also introduces one to two orders of magnitude of computational overhead compared to the other methods (Sundararajan et al. [34] recommend 20–300 gradient evaluations to approximate attributions). For the $\mathcal{X}$-AlexNet, however, our $\mathcal{X}$-Gradient method is on par with Integrated Gradients and produces the best attributions while requiring only one gradient evaluation, and therefore, a fraction of the compute power. Since the input gradient and Expected Gradients [7] with only one reference sample do not satisfy many of the desirable axioms (see Table 1), they produce clearly lower quality attributions as expected. Note that high-qualitative attribution methods should perform well across all the listed metrics, which is why the input gradient is not a competitive attribution method even though it performs well on the RAM metric. To conclude, we can see that our $\mathcal{X}$-Gradient attribution yields a significant improvement in quality compared to state-of-the-art generic attribution methods that require similar computational cost. This suggests that our effort to produce an efficient and high-quality attribution method is justified and accomplished.

## 4.3 Training with attribution priors

To benchmark our approach against other attribution methods when training with attribution priors, we replicate the sparsity experiment introduced in [7]. To that end, we employ the public NHANES I survey data [17] of the CDC of the United States, containing 118 one-hot encoded medical attributes, *e.g.*, age, sex, and vital sign measurements, from 13,000 human subjects (no personally identifiable information). The objective of the binary classification task is to predict if a human subject will be dead (0) or alive (1) ten years after the data was measured. A simple MLP with ReLU activations is used as the model. Therefore, it can be transformed into an $\mathcal{X}$-DNN by simply removing the bias terms. To emulate a setting of scarce training data and to average out variance, we randomly

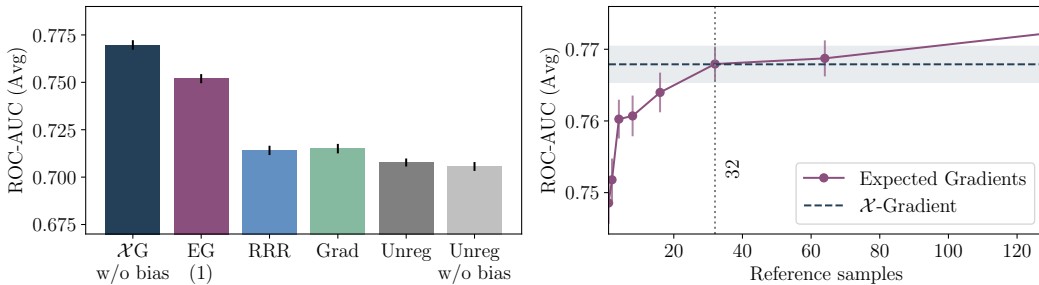

Figure 1: (left) *Average ROC-AUC* across 200 randomly subsampled datasets for the same attribution prior using different attribution methods. "w/o bias" denotes that the bias term has been removed from the MLP. (right) *Average ROC-AUC* across 200 randomly subsampled datasets of Expected Gradients (EG) over the number of reference samples. The current state-of-the-art EG requires approximately 32 reference samples, and thus, 32 times more computational power to outmatch $\mathcal{X}$G. Confidence intervals indicate two times the standard error of the mean.

subsample 200 training and validation datasets containing 100 data points from the original dataset. Erion et al. [7] proposed a novel attribution prior that maximizes the Gini coefficient, *i.e.*, minimizes the statistical dispersion, of the feature attributions. They show that this allows to learn sparser models, which have improved generalizability on small training datasets. The more faithfully the attribution reflects the true behavior of the model, the more effective the attribution prior should be.

**Comparing attribution methods.** We compare different attribution methods that have previously been used for training with attribution priors and require only one gradient evaluation; thus, they have comparable computational cost. The results in Fig. 1(left) show that our method ($\mathcal{X}$G w/o bias) outperforms all other competing methods. We can also see that for the unregularized model, removing the bias (Unreg w/o bias) has almost no effect on the average ROC-AUC of the method, once again showing that our modification for making attributions efficient, *i.e.*, removing the bias term, is plausible in various scenarios.

Since the attribution quality of Expected Gradients can be improved using more reference samples, as this yields a better approximation to the true integral, we plot the average ROC-AUC of Expected Gradients over the number of reference samples used in Fig. 1(right). We can clearly see that adding more samples improves the ROC-AUC when training with an EG attribution in the prior, yet again, showing that higher quality attributions lead to more effective attribution priors [7]. However, we also find that approximately 32 reference samples are needed, and hence 32 times more computational power, to match the quality of our efficient $\mathcal{X}$-Gradient method. When using more than 32 reference samples, Expected Gradients slightly outperform our method in terms of ROC-AUC, which is due to the limitations discussed in Sec. 3 (fixed baseline, no bias terms). We argue that it is often worth accepting this small accuracy disadvantage in light of the significant gain in efficiency of computing high-quality feature attributions.

To put this improvement in efficiency into perspective, we measure the computation time of training a ResNet-50 on the ImageNet dataset when using Expected Gradients with 32 reference samples and $\mathcal{X}$-Gradient (see supplemental material). Using a single GPU, the computational overhead introduced when using Expected Gradients with 32 reference samples amounts to an approximately 130-fold increase in the required computation time compared to training with $\mathcal{X}$-Gradient, and thus, would turn several days of training into several months of training.

### 4.4 Homogeneity of $\mathcal{X}$-DNNs

The fundamental difference between $\mathcal{X}$-DNNs and regular DNNs is the nonnegative homogeneity of the former. To show implications on the model and its attributions, we conduct the following experiment. Similarly to Hendrycks and Dietterich [10], we reduce the contrast of the ImageNet [24] validation split by multiplying each image with varying factors $\alpha$ and report the top-1 accuracy of AlexNet and the corresponding $\mathcal{X}$-AlexNet from Sec. 4.1, *i.e.*, they have not been trained specifically to handle contrast changes. Results can be found in Fig. 2(left). We can observe that decreasing the image contrast leads to a strong drop in accuracy of a regular AlexNet. On the other hand, due to the

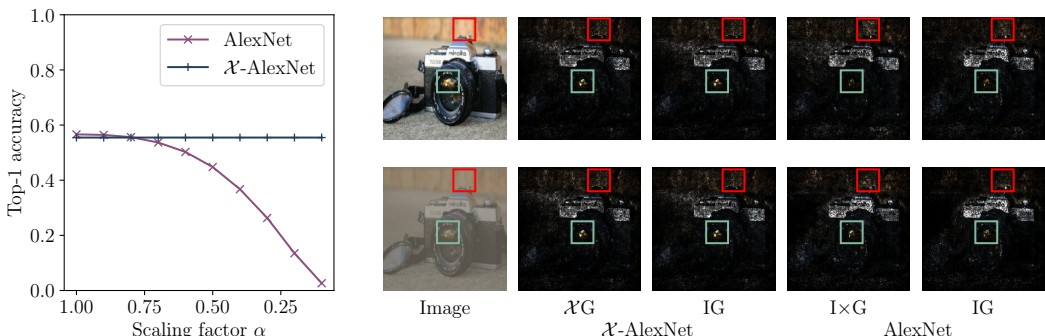

Figure 2: (left) *Top-1 accuracy* for AlexNet and $\mathcal{X}$-AlexNet on the ImageNet validation split with decreasing contrast (scaled by $\alpha$). Due to the nonnegative homogeneity of $\mathcal{X}$-AlexNet, the accuracy does not drop when reducing the contrast. (right) *Qualitative examples* of normalized attributions for $\mathcal{X}$-AlexNet and AlexNet using the attribution methods $\mathcal{X}$-Gradient ($\mathcal{X}$G) resp. Input×Gradient (I×G) as well as Integrated Gradients (IG). The displayed attributions obtained from $\mathcal{X}$-AlexNet are almost identical, while attributions obtained from AlexNet differ significantly (see highlighted areas).

equivariance to contrast of $\mathcal{X}$-DNNs, the accuracy of the $\mathcal{X}$-AlexNet is unaffected, showing improved robustness towards multiplicative contrast changes. To give some qualitative examples, in Fig. 2(right) we plot the attributions for the output logit of the target class ('reflex camera') for a regular AlexNet and an $\mathcal{X}$-AlexNet for an original image and the corresponding low-contrast image obtained by multiplying the normalized image with $\alpha = 0.3$. For the $\mathcal{X}$-AlexNet, our $\mathcal{X}$-Gradient method and Integrated Gradients (IG) [34] produce attributions that are identical up to a small approximation error; reducing the image contrast keeps the attributions unchanged up to a scaling factor (not visible due to normalization for display purposes). However, the displayed attributions from the regular AlexNet differ significantly between Input×Gradient (I×G) and IG, as well as for different contrasts (see highlighted areas). We argue that the above properties of $\mathcal{X}$-DNNs generally reflect desirable properties and show that they behave more predictably with contrast changes than regular DNNs. Also note how high feature attribution scores can arise in the background (*e.g.*, red box), showing how DNN predictions can depend of parts of the input that do not appear salient for the category; this highlights a possible use case for attribution priors [22, 23] enabled by our approach.

## 5 Conclusion and broader impact

In this work, we consider a special class of efficiently axiomatically attributable DNNs, for which an axiomatic feature attribution can be computed in a single forward/backward pass. We show that nonnegatively homogeneous DNNs, termed $\mathcal{X}$-DNNs, are efficiently axiomatically attributable and establish a new theoretical connection between Input×Gradient [27] and Integrated Gradients [34] for nonnegatively homogeneous DNNs of different degrees. Moreover, we show that many commonly used architectures can be transformed into $\mathcal{X}$-DNNs by simply removing the bias term of each layer, which has a surprisingly minor impact on the accuracy of the model in two application domains. The resulting efficiently computable and high-quality feature attributions are particularly well-suited for inclusion into the training process and potentially enable a wide application of axiomatic attribution priors. This could, for example, be used to reliably mitigate dependence on unwanted features and biases induced by the training dataset, which is a major challenge in today's ML systems.

Obermeyer et al. [20] found evidence that a widely used algorithm in the U.S. health care system contains racial biases that can be traced back to biases in the dataset that was used to develop the algorithm. Using our method to generate high-quality attributions that reflect the true behavior of an $\mathcal{X}$-DNN and an appropriate attribution prior, such problems could potentially be resolved (though more research on attribution priors is necessary). However, allowing biases to be controlled by an ML practitioner can introduce new risks. Just like datasets, humans are also not free of biases [35], which can potentially be reflected in such priors. We as a society need to be careful that this responsibility is not exploited and used for discriminatory or harmful purposes. One way to approach this problem for applications that affect the general public is to introduce an ethical review committee, which assesses whether the proposed priors are legitimate or reprehensible.

## Acknowledgments and disclosure of funding

We would like to thank Jannik Schmitt for helpful mathematical advice. This project has received funding from the European Research Council (ERC) under the European Union's Horizon 2020 research and innovation programme (grant agreement No. 866008). The project has also been supported in part by the State of Hesse through the cluster projects "The Third Wave of Artificial Intelligence (3AI)" and "The Adaptive Mind (TAM)".

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
