# Fast Axiomatic Attribution for Neural Networks
## – Supplemental Material –

**Robin Hesse[1]**          **Simone Schaub-Meyer[1]**          **Stefan Roth[1,2]**

[1]Department of Computer Science, TU Darmstadt     [2]hessian.AI
{robin.hesse, simone.schaub, stefan.roth}@visinf.tu-darmstadt.de

## A   Proofs and further results

*Proof details for Proposition 3.2.* In the proof of Proposition 3.2, we make use of the property that the derivative of a $k^{\text{th}}$-order homogeneous and differentiable function $F$ is a $(k-1)^{\text{st}}$-order homogeneous function, *i.e.*,

$$\frac{\partial F(\alpha x)}{\partial \alpha x_i} = \alpha^{k-1} \frac{\partial F(x)}{\partial x_i}, \tag{9}$$

see, *e.g.*, Corollary 4 in [43]. Assuming $k^{\text{th}}$-order homogeneity of $F$ and using the chain-rule, the above Eq. (9) follows from

$$\alpha \frac{\partial F(\alpha x)}{\partial \alpha x_i} = \frac{\partial F(\alpha x)}{\partial \alpha x_i} \frac{\partial \alpha x_i}{\partial x_i} = \frac{\partial F(\alpha x)}{\partial x_i} = \frac{\partial \alpha^k F(x)}{\partial x_i} = \alpha^k \frac{\partial F(x)}{\partial x_i}. \tag{10}$$

$\square$

*Proof of Proposition 3.8.* For any input $z \in \mathbb{R}^n$, $\alpha \in \mathbb{R}_{\geq 0}$, and piecewise linear activation function $\phi_l$ according to Eq. (7), we want to show that

$$\alpha \phi_l(z) = \phi_l(\alpha z). \tag{11}$$

For $\alpha = 0$, both sides evaluate to 0 and the equality holds. For $\alpha > 0$, the equality holds as long as the active interval of the activation function does not change. The active interval changes either when the sign of the input is changed, *i.e.*, it goes from positive to negative or vice versa, or when a positive input changes to 0, or a value of 0 changes to positive. Since $\alpha > 0$, a multiplication with $\alpha$ can neither change the sign nor can make positive values 0 or 0 values positive. Therefore, scaling the input with $\alpha > 0$ changes none of the active activation function intervals and nonnegative homogeneity for $\alpha \in \mathbb{R}_{\geq 0}$ holds. $\square$

*Proof of Proposition 3.9.* For any input $z \in \mathbb{R}^n$, $\alpha \in \mathbb{R}_{\geq 0}$, and pooling function $\psi_l$ with the assumed properties, we want to show that

$$\alpha \psi_l(z) = \psi_l(\alpha z). \tag{12}$$

If the pooling function is linear, homogeneity implicitly holds. If the pooling function is selecting values based on their relative ordering, we consider two cases. For $\alpha = 0$, both sides evaluate to 0 and the equality holds. For $\alpha > 0$, the relative ordering of the entries in $z$ is unchanged by a scaling with $\alpha$, hence the same entry is selected by the pooling function. Since the value of the selected entry is scaled by $\alpha$, the above Eq. (12) holds for $\alpha \in \mathbb{R}_{\geq 0}$ and nonnegative homogeneity is satisfied. $\square$

*Proof that $\mathcal{X}$-Gradient satisfies nonnegative homogeneity (Definition 3.6).* Using Eq. (9) and nonnegative $1^{\text{st}}$-order homogeneity of any $\mathcal{X}$-DNN $F$, it follows that

$$\mathcal{X}\text{G}(F, \alpha x) = \alpha x_i \frac{\partial F(\alpha x)}{\partial \alpha x_i} = \alpha x_i \alpha^0 \frac{\partial F(x)}{\partial x_i} = \alpha \mathcal{X}\text{G}(F, x), \tag{13}$$

for $\alpha \in \mathbb{R}_{\geq 0}$, and therefore, nonnegative homogeneity of the attribution is satisfied. $\square$

35th Conference on Neural Information Processing Systems (NeurIPS 2021).

**Axiomatic attributions.** Table 1 in the main text summarizes the axioms [34] that are satisfied by several attribution methods. For proofs of the axioms that are satisfied by Integrated Gradients, please refer to [34]. For proofs of the axioms that are satisfied by Expected Gradients, please refer to [7]. For proofs of the axioms that are satisfied by Input×Gradient and Gradient, please refer to [7, 34] and see below. As $\mathcal{X}$-Gradient equals Integrated Gradients for $\mathcal{X}$-DNNs according to Proposition 3.2, all the axioms satisfied by Integrated Gradients are also satisfied by $\mathcal{X}$-Gradient (for $\mathcal{X}$-DNNs).

For Expected Gradients to satisfy the same axioms that are satisfied by Integrated Gradients, convergence must have occurred, which can only be expected after multiple gradient evaluations. To emphasize the advantage of our method when only considering attribution methods that use a single gradient evaluation, in Table 1 we also show the axioms that are satisfied by Expected Gradients [7] when using only one reference sample, *i.e.*, when convergence did not yet occur. Proof sketches for the axioms satisfied by Expected Gradients with only one reference sample are as follows:

1. *Sensitivity (a):* Since there exist networks for which Sensitivity (a) is not satisfied by Input×Gradient, and Expected Gradients could choose a sample such that the approximation equals Input×Gradient, Sensitivity (a) is also not satisfied by Expected Gradients in general.

2. *Sensitivity (b):* As the gradient *w.r.t.* an irrelevant feature will always be zero, Sensitivity (b) is satisfied.

3. *Implementation invariance:* As Expected Gradients use stochastic sampling for the baseline, there is no guarantee that even for the same model two attributions are equal.

4. *Completeness:* Again, following the argument from Sensitivity (a), Completeness is not given.

5. *Linearity:* As Expected Gradients use a stochastic sampling for the baseline, there is no guarantee that Linearity holds.

6. *Symmetry-preserving:* Following the argument from Linearity, Symmetry-preserving does not hold.

**Why is nonnegative homogeneity a desirable axiom for attribution methods?** Explainability is closely related to predictability. Knowing how a model behaves under certain changes to the input implies an understanding of the model. Therefore, axioms like *linearity* [34] and *nonnegative homogeneity*, which essentially describe a form of predictability, are generally desirable and allow for a more complete understanding of the model's behavior.

**(Input×)Gradient violates Sensitivity (a).** To see that gradients and Input×Gradient violate Sensitivity (a), it is instructive to consider the concrete example given in [34]: Assume we have a simple ReLU network $f(x) = 1 - \text{ReLU}(1-x)$. When having a baseline $x' = 0$ and an input $x = 2$, $f(x')$ respectively $f(x)$ changes from 0 to 1. However, as the function flattens out at $x = 1$, the above gradient-based attribution methods would yield an attribution of 0 for the input $x = 2$.

# B   Experimental details

In the following section we provide additional details to ensure reproducibility of our experiments. For further information, please see our public code base[1] released under an Apache License 2.0.

## B.1   Removing the bias term in DNNs

The models for all reported results in Sec. 4.1 have been trained for 100 epochs on the training split of the ImageNet [24] dataset with a batch size of 256 and using a single Nvidia A100 SXM4 (40GB) GPU. The training time per epoch is approximately 10 minutes for AlexNet, 60 minutes for VGG16, and 40 minutes for ResNet-50. For training the AlexNet and VGG models, we use the official PyTorch [44] implementation[2] that is published under a BSD 3-Clause license. We use an SGD optimizer with an initial learning rate of 0.01 that is decayed by a factor of 0.1 every 30 epochs, a momentum of 0.9, and a weight decay of 1e-4. For training the ResNet models, we use the settings proposed by [40] and rely on the publicly available code,[3] which is released under a BSD 3-Clause

---

[1]github.com/visinf/fast-axiomatic-attribution
[2]github.com/pytorch/examples
[3]github.com/hongyi-zhang/Fixup

license. The hyperparameters are the same as for the AlexNet and VGG models except that we use mixup regularization [45] with an interpolation strength $\alpha = 0.7$, a cosine annealing learning rate scheduler, and an initial learning rate of 0.1.

The mean absolute relative difference between the attribution obtained from Integrated Gradients [34] and the attribution obtained from calculating Input×Gradient for regular DNNs resp. $\mathcal{X}$-Gradient for $\mathcal{X}$-DNNs is calculated as

$$d(\mathcal{A}, X) = \frac{1}{n|X|} \sum_{x \in X} \sum_{i=1}^{n} \frac{|\mathrm{IG}_i(F, x, \mathbf{0}) - \mathcal{A}_i(F, x)|}{|\mathrm{IG}_i(F, x, \mathbf{0})|} , \tag{14}$$

with $X$ denoting a dataset consisting of samples $x \in \mathbb{R}^n$.

## B.2 Benchmarking gradient-based attribution methods

For the experimental comparison of gradient-based attribution methods in Sec. 4.2, we use the models from Sec. 4.1 (see Appendix B.1 for details) and evaluate on the ImageNet validation split using a single Nvidia A100 SXM4 (40GB) GPU. To quantify the quality of attributions, we use the attribution quality metrics proposed by Lundberg et al. [15]. The metrics reflect how well an attribution method captures the relative importance of features by masking out a progressively increasing fraction of the features based on their relative importance:

**Keep Positive Mask (KPM)** measures the attribution method's capability to find the features that lead to the greatest increase in the model's output logit of the target class. For that a progressively increasing fraction of the features is masked out, ordered by least positive to most positive attribution. Then the AUC of the resulting curve is measured. Intuitively, if an attribution reflects the true behavior of the model, unimportant features will be masked out first and the model output logit decreases only marginally, resulting in a high value for the AUC. The other way around, when an attribution does not reflect the true behavior of the model, an important feature might be masked out too early and the target class output decreases quickly, leading to a smaller score.

**Keep Negative Mask (KNM)** works analogously for negative features. This means that the better the attribution, the smaller the metric. Note that for KPM and KNM, all negative and positive features are masked out by default, respectively.

**Keep Absolute Mask (KAM)** and **Remove Absolute Mask (RAM)** work similarly but using the absolute value of the attributions and measuring the AUC of the top-1 accuracy. For KAM, we keep the most important features and measure the AUC of the top-1 accuracy over different fractions of masking. A high-quality attribution method should keep the features most important for making a correct classification, and therefore, the metric should be as high as possible. RAM masks out the most important features first, meaning that the accuracy should drop fast. Therefore, a smaller value indicates a better attribution.

As we evaluate attributions for image classification models, we adapt the above metrics to work with image data. This is achieved by replacing the masked pixels with those of a blurry image, which is obtained using a Gaussian blur with a kernel size of $51 \times 51$ and $\sigma = 41$ applied to the original input image. The parameters were chosen such that the resulting image is visually heavily blurred. This ensures that features can properly be removed.

## B.3 Training with attribution priors

Our experiment with attribution priors in Sec. 4.3 replicates the experimental setup of [7]. We use the original code, which includes the NHANES I dataset and is published under the MIT license.[4] We use the attribution prior proposed by Erion et al. [7] to learn sparser models, which have improved generalizability. The prior is defined as

$$\Omega_{sparse}(\bar{\mathcal{A}}) = -\frac{\sum_{i=1}^{n} \sum_{j=1}^{n} |\bar{\mathcal{A}}_i - \bar{\mathcal{A}}_j|}{m \sum_{i=1}^{n} \bar{\mathcal{A}}_i} ,$$

---

[4]`github.com/suinleelab/attributionpriors`

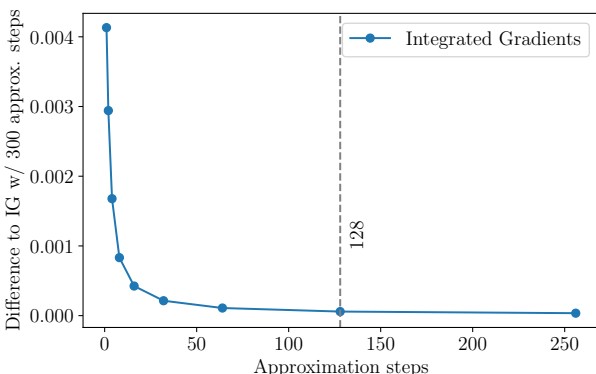

Figure 3: *Convergence of Integrated Gradients [34].* We plot the mean absolute difference of Integrated Gradients obtained by using 300 and different numbers of approximation steps for AlexNet on the ImageNet validation split. We find convergence to occur after approximately 128 steps.

with $\bar{A}$ denoting the mean attribution of a mini-batch with $m$ samples. This prior improves sparsity of the model by minimizing the statistical dispersion of the feature attributions. We use the following attribution methods as baselines, which are commonly used for training with attribution priors: Expected Gradients (EG), the input gradient of the *log* of the output logit as proposed by [23] (RRR), and a regular input gradient (Grad). We compare these methods with our novel $\mathcal{X}$-Gradient ($\mathcal{X}$G) attribution method. For each attribution method, we perform an individual hyperparameter search to find the optimal regularization strength $\lambda \in \{0.01, 0.1, 1, 10, 100\}$. We find $\lambda = 0.1$ for RRR and Grad, and $\lambda = 1.0$ for $\mathcal{X}$G and EG. When training the model with Expected Gradients using more than one reference sample, we continue to use the regularization strength $\lambda$ that was found using one reference sample. All other hyperparameters are kept as in the original experiment of [7]. To train the models, we use a Nvidia GeForce RTX 3090 (24GB) GPU.

To provide a numerical comparison of the efficiency of $\mathcal{X}$-Gradient and Expected Gradients [7], we report the computation time and GPU memory usage for training a ResNet-50 on the ImageNet dataset with Expected Gradients using 32 reference samples and with $\mathcal{X}$-Gradient. We use a single Nvidia A100 SXM4 (40GB) GPU and a batch size of two. The number of reference samples corresponds to the number of reference samples determined in the experiment in Fig. 1(right), where both networks achieve the same ROC-AUC. When using Expected Gradients for training, the GPU memory usage is 14.57 GB while for $\mathcal{X}$-Gradient the memory usage is 4.21 GB. The computation time per iteration, averaged over 100 iterations, is 1.12 s for Expected Gradients and 0.0086 for $\mathcal{X}$-Gradient. To conclude, in this scenario we observe a massive improvement in the efficiency of $\mathcal{X}$-Gradient compared to Expected Gradients. Expected Gradients requires $\sim 130$ times more computation time and $\sim 3.46$ times more GPU memory.

### B.4 Homogeneity of $\mathcal{X}$-DNNs

For the experiment in Sec. 4.4, we use the same models as in Sec. 4.1 (see Appendix B.1 for details) and evaluate on the ImageNet validation split using a single Nvidia A100 SXM4 (40GB) GPU. Additional qualitative examples of the attributions for the output logit of the target class for a regular AlexNet and an $\mathcal{X}$-AlexNet, as in Fig. 2(right), are shown in Fig. 4. Our findings in Sec. 4.4 are consistent with the additional results. As with Fig. 2(right) in the main paper, we observe that $\mathcal{X}$-Gradient ($\mathcal{X}$G) equals Integrated Gradients (IG) for the $\mathcal{X}$-AlexNet up to a small approximation error and that reducing the contrast of the images keeps the attribution unchanged up to a scaling factor (not visible due to normalization for display purposes). On the other hand, for the regular AlexNet the attributions obtained from Input×Gradient and Integrated Gradients differ and change depending on the contrast.

### B.5 Convergence of Integrated Gradients

For our experimental comparisons with Integrated Gradients [34], we assume convergence of the method. To empirically find a suitable number of approximation steps, we analyze the mean absolute

difference of the Integrated Gradients obtained by using $n$ and 300 approximation steps as plotted in Fig. 3. We choose 300 steps for reference because Sundararajan et al. [34] report 300 steps as the upper bound for convergence. We use the trained AlexNet model from Sec. 4.1 (details in Appendix B.1), the ImageNet validation split, and $n \in [1, 2, 4, 8, 16, 32, 64, 128, 256]$. We find 128 approximation steps to be sufficient and use this number in our experiments.

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

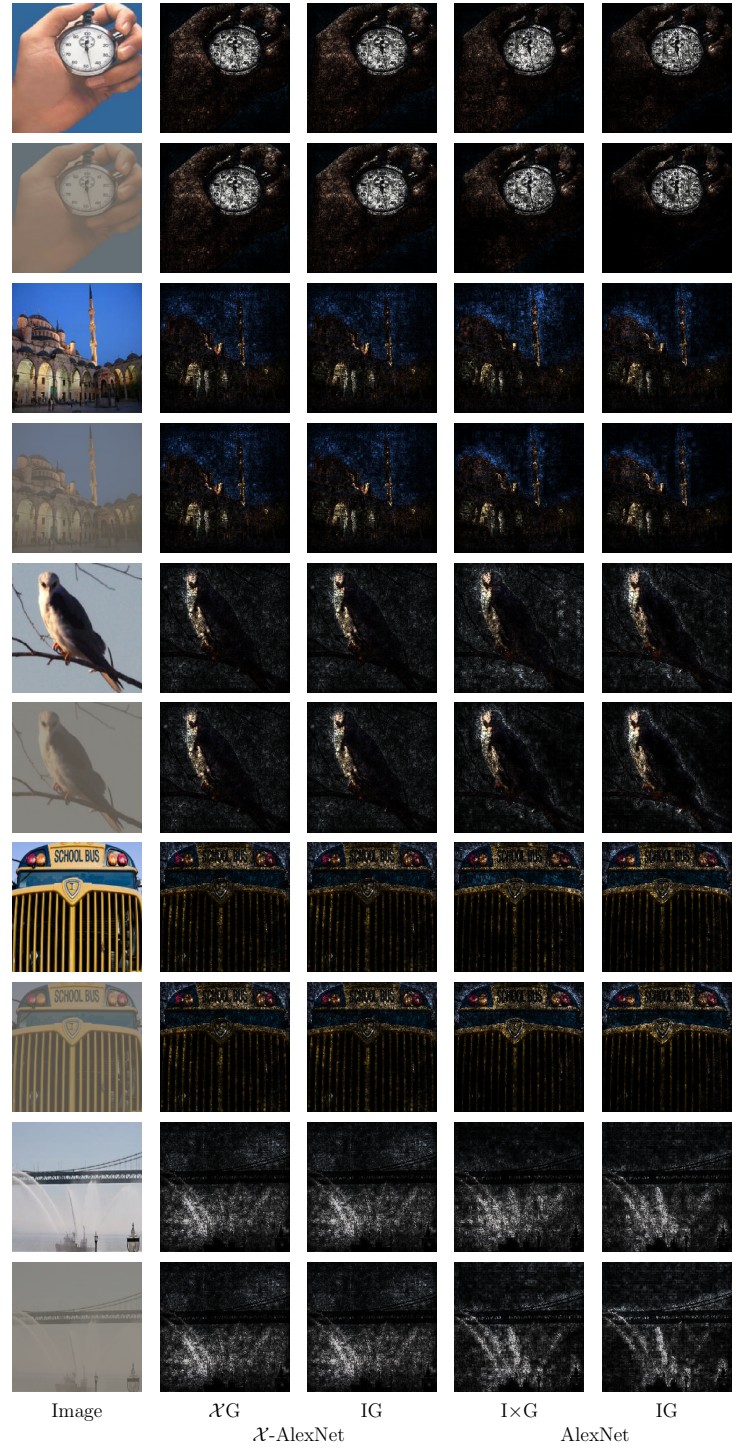

Figure 4: *Qualitative examples* of normalized attributions for the output logit of the target class for $\mathcal{X}$-AlexNet and AlexNet using the attribution methods Input×Gradient (I×G), $\mathcal{X}$-Gradient ($\mathcal{X}$G), and Integrated Gradients (IG).