# OpenReview forum: "Fast Axiomatic Attribution for Neural Networks"
_NeurIPS.cc/2021/Conference — NeurIPS 2021 Poster_

### Official Review · Reviewer_wVX5 · 2021-06-24

**Rating:** 5
**Confidence:** 3

**Summary:**

This paper proposes to use non-negatively homogeneous DNN to perform feature attribution with better computational efficiency at a minimal performance cost. The so-called non-negatively homogeneous DNN is obtained by removing the bias term from most common neural networks and using a Relu/Leaky_Relu activation function. By doing this, the model would be linearly scaled by the input data (F(ax) = a*F(x)). With this property, the integrated Gradient (IG) method can be easily computed with one call of gradient computing, which improves its original computational efficiency. In the experiment, this model shows comparable performance with the IG method but much less computation effort.



**Main Review:**

I’m not an expert in feature attribution research but this paper sounds like an interesting discovery based on the existing IG method, which takes advantage of neural networks that are non-negatively homogeneous after removing their "Bias" terms. However, the key idea of this paper that improves the IG method looks very simple and straightforward (for myself, I know the non-negatively homogeneity of DNNs without bias term a long time ago from many papers discussing the property of DNNs. e.g., for papers like "Analytic Marching: An Analytic Meshing Solution from Deep Implicit Surface Networks", they already kind of show these results by analyzing the local linearity of the neural networks). At least, this conclusion is not very novel from my perspective. On the other hand, this discovery could be an interesting application of this "property" on the feature attribution research. Therefore, I think the paper should give a better introduction to the background of this problem (feature attribution) and demonstrate that there is enough contribution based on the efficiency improvement of the IG method. For example, is "IG" the state-of-the-art method for feature attribution while being limited by its computational efficiency? Is there any other feature attribution method that are efficient compared with one-gradient-call IG?

Minor questions:
1. for clarity, can the author provide a more solid example of what the Integrated Gradients method really does?
2. What is the relative attribution distance? (Mentioned in Table 2)


**Time Spent Reviewing:**

2

---

> ### Author Response · Authors · 2021-08-10
> **Author response to reviewer wVX5**
>
> We thank the reviewer for the detailed feedback and valuable suggestions to further improve our work. We hope to adequately address the reviewer's concerns and that we are able to clarify the significance of our main contributions in points 1 and 2. We will ensure that the revision becomes sufficiently self-explaining to reliably deliver the ideas of our work.
>
> > **1.** _"However, the key idea of this paper that improves the IG method looks very simple and straightforward (for myself, I know the non-negatively homogeneity of DNNs without bias term a long time ago from many papers discussing the property of DNNs."_
>
> We agree that the finding that removing the bias from specific DNNs results in nonnegative homogeneity is an already known fact and will include more appropriate references to the revision. However, we do not see this finding as a key idea of our paper. Instead, we find an efficiently computable closed-form solution of Integrated Gradients for a class of DNNs that has a similar capacity as regular DNNs, namely strictly positive homogeneous DNNs of degree k (with $k \geq 1$). We consider a concrete instantiation of this class of models (nonnegatively homogeneous DNNs without bias are just one example, there might be others) that is particularly well suited for training with attribution priors. In two experiments, we demonstrate that this class indeed has almost the same capacity as regular networks while being able to efficiently compute high-quality attributions. This allows us to beat state-of-the-art methods for training with attribution priors while requiring only a fraction of the computing power (Section 4.3). The significance of this contribution is further acknowledged by reviewers gwR5 and bst1.
>
> > **2.** _"Therefore, I think the paper should [...] demonstrate that there is enough contribution based on the efficiency improvement of the IG method."_
>
> Besides our theoretical contributions, we and reviewers gwR5 and bst1 believe that there is significant practical contribution based on the efficiency improvement. In our experiments as well as in prior work [7, 13], training DNNs with axiomatic attributions (IG [31], EG [7]) is considered advantageous on the predictive performance side because they produce high-quality attributions. However, they increase the training time significantly. As far as we are aware, we are the first to be able to break this tradeoff by considering nonnegatively homogeneous DNNs for which we can compute the Integrated Gradient with only one gradient evaluation. In our practical example in lines 328-330, this allows us to reduce the run time by a large factor of 130 while reducing the used GPU memory by a factor of 3.46 (see reviewer gwR5, comment 5).
>
> > **3.** _"Therefore, I think the paper should give a better introduction to the background of this problem (feature attribution) [...]."_
>
> We thank the reviewer for the valuable suggestions and we will adequately address the listed questions in the revision to give a better introduction to the background of feature attribution.
>
> > **4.** _"For example, is "IG" the state-of-the-art method for feature attribution while being limited by its computational efficiency?"_
>
> As it is hard to evaluate attribution methods and it depends on the application, there is no real state of the art. However, Integrated Gradients provably satisfies some desirable axioms that are generally not satisfied by other attribution methods and often showed superior performance compared to other more efficient gradient-based approaches (see Section 4.2.). Additionally, Integrated Gradients is a highly impactful work that has been studied in recent works on attribution priors [7,13].
>
> > **5.** _"Is there any other feature attribution method that are efficient compared with one-gradient-call IG?"_
>
> Currently, there is a tradeoff between the efficiency of a method and the quality of the produced attributions. There are methods like the saliency method [28] and Input x Gradient [25] that require only one gradient call but produce attributions that are generally of lower quality compared to Integrated Gradients. Integrated Gradients, on the other hand, produces high-quality attributions but requires multiple gradient evaluations (20-300) (see Section 4.2). To the best of our knowledge, our work is the first to combine the high quality of Integrated Gradients with the efficiency of Input x Gradient for DNNs.
>
> > **6.** _"for clarity, can the author provide a more solid example of what the Integrated Gradients method really does?"_
>
> For a detailed explanation of the Integrated Gradients method, we cited the original paper by Sundararajan et al. [31]. Additionally, we will clarify the usage and motivation of Integrated Gradient in the revised paper.
>
> > **7.** _"What is the relative attribution distance? (Mentioned in Table 2)"_
>
> As described in Section 4.1. and in the supplemental material, the relative distance in Table 2 is the mean relative distance between the attribution obtained from Integrated Gradients and the attribution obtained from calculating Input x Gradient for regular DNNs resp. $\mathcal{X}$-Gradient for $\mathcal{X}$-DNNs over the ImageNet validation split. We agree that Table 2 could be more self-explaining and will add appropriate references.
>
> We hope that we have been able to adequately clarify the reviewer's concerns and look forward to further discussions.

---

### Official Review · Reviewer_gwR5 · 2021-06-26

**Rating:** 7
**Confidence:** 4

**Summary:**

The paper first identifies the intensive computational cost in using Integrated Gradient (IG) as an attribution prior in training explainable deep neural networks. To this end the authors propose to switch the network architecture to negatively homogeneous networks, specifically networks without bias term, a.k.a. $\mathcal{X}$-DNN, in this paper, where they prove that $\texttt{input} \times \texttt{gradient}$ is an equivalent attribution method to IG; and therefore can decrease the number of gradient evaluations in computing IG. The paper then conducts several experiments over the ImageNet dataset and several tabular datasets to validate the performance of the proposed method.

**Limitations And Societal Impact:**

The authors adequately addressed the limitations and potential negative societal impact of their work.

**Main Review:**

In general, I find the theoretical results are very interesting and it opens another fresh view of connecting Integrated Gradient with inherent properties of networks, just like the connection between randomized smoothing and SmoothGrad. However, the evaluation part of the paper is confusing to me, which prevents me from giving higher scores. However, I am on the fence and expect the authors’ response to clear my concerns. Please find the detail as follows.

### Originality
The work is a novel combination of well-known techniques. The related work is well-cited.

### Quality:
I have two major concerns on the quality side of this paper. Please find the detail as follows:
1) Theoretical limitations of no-bias network, a.k.a.$ \mathcal{X}$-DNN. The authors provide empirical evidence that $\mathcal{X}$-DNNs match the Top-5 accuracy of regular networks in Section 4.1; however the experiment is only limited to one data distribution and one network, which is far away from covering all kinds of applications the author believes their work can potentially benefit for, e.g. health care. I would suggest that more theoretical analysis can be provided to convince the reader $\mathcal{X}$-DNNs are as useful as regular networks. An example question is: assuming zero vectors as baselines can decourage the network to use features with values close to 0. For example, it will always consider darker pixels as not useful so black dogs may always have smaller attribution values compared to white dogs. Will the $\mathcal{X}$-DNN end up with becoming a network that is always more sensitive to “bright” features and less sensitive to “darker” features that are actually important?

2) Evaluations are not well-motivated. My biggest concern is that several proposed evaluations and their metrics are not well-motivated to show the proposed network architecture is promising.
- Firstly, as$\mathcal{X}$G is not a new method but more like an alias of input x grad for networks without bias and is mathematically equivalent to IG in the proposed architecture, the result of Table 3 is actually comparing existing attributions using existing metrics (KPM, KNM, KAM and RAM). More analysis could be added to justify why Table 3 is important to include and why the result is useful to show the proposed $\mathcal{X}$-DNN is promising.  Besides, the motivations of using metrics from [1] instead of other work, e.g. [2], are unclear to the reader.

- Secondly, the paper is motivated to accelerate the training with attribution priors but statistics of the timing and memory cost in training $\mathcal{X}$-DNN are missing in Section 4. More numerical results instead of descriptions, e.g. line 328-330, will show exactly how fast and efficient the proposed algorithm is.

[1] S. M. Lundberg, G. Erion, H. Chen, A. DeGrave, J. M. Prutkin, B. Nair, R. Katz, J. Himmelfarb, N. Bansal, and S.-I. Lee. From local explanations to global understanding with explainable AI for trees. Nature Machine Intelligence.

[2] Yeh, C., Hsieh, C., Suggala, A.S., Inouye, D.I., & Ravikumar, P. (2019). On the (In)fidelity and Sensitivity of Explanations. NeurIPS.

### Clarity
The motivation is clearly stated: the attribution prior regularization is usually computational expensive in terms of the number of gradient evaluations w.r.t the input; therefore, it is naturally to find another way of computing attributions with less number of gradient evaluations while the attribution used in the training is also faithful;
2) the paper describes the motivation and the formulation of \mathcal{X}-DNN clearly through multiple definitions and propositions. The equations and proofs are easy to understand.

### Significance
The results are important and others (researchers or practitioners) are likely to use the ideas if the technical questions can be resolved accordingly.

### Update from discussion
Thanks for the authors to post the thoughtful responses to my questions. Even though some questions are probably not able to resolve during this discussion time, e.g. the theoretical guarantees about the bias-free networks, I generally find this paper to be interesting and the authors' response is convincing and they are trying to make the changes that can answer my questions in the future revision. I therefore will increase my score from 5 to 7.

**Time Spent Reviewing:**

7

---

> ### Author Response · Authors · 2021-08-10
> **Author response to Reviewer gwR5**
>
> We thank the reviewer for the detailed feedback, for pointing out the strengths of our work, and for valuable suggestions to further improve our work.
>
> > **1.** _"The authors provide empirical evidence that $\mathcal{X}$-DNNs match the Top-5 accuracy of regular networks in Section 4.1; however the experiment is only limited to one data distribution and one network, which is far away from covering all kinds of applications the author believes their work can potentially benefit for, e.g. health care."_
>
> We would like to politely point out that our experiments are not limited to only one data distribution and one network. In Section 4.1 we evaluate three different $\mathcal{X}$-DNNs ($\mathcal{X}$-AlexNet, $\mathcal{X}$-VGG16, and $\mathcal{X}$-ResNet50) on the challenging ImageNet dataset. Further, in Section 4.3, we evaluate a multilayer perceptron with and without bias on tabular health data (Fig 1 _Unreg_ vs. _Unreg w/o bias_). We will clarify that for Figure 1. Nonetheless, we agree that it is impossible to empirically show evidence that removing the bias is always feasible. For this reason and in accordance with reviewer bst1, we will include suggestions for practical usage such as a preliminary analysis of the bias-less network to estimate its predictive performance before training with attribution priors. Generally, as shown in Section 4.3. it always comes down to a tradeoff between maximum accuracy and efficiency. So the question of whether our method is useful depends on the availability of resources and how much of a drop in accuracy is to be expected when removing the bias.
>
> > **2.** _"I would suggest that more theoretical analysis can be provided to convince the reader $\mathcal{X}$-DNNs are as useful as regular networks."_
>
> As it is extremely hard to theoretically show that no-bias DNNs will always be almost on par with regular DNNs, we build our reasoning on empirical evidence and simple practical suggestions, e.g. preliminary analysis (see point 1). Indeed, prior theoretical work on the role of biases in DNNs [33] suggests that the bias is important for predictive performance, which is in contrast to our empirical findings. We hypothesize that when removing the bias term, the DNN learns some kind of layer averaging strategy that compensates for the missing bias. One noteworthy theoretical limitation that follows from the nonnegative homogeneity of no-bias DNNs $F$, is that $F(\mathbf{0}) = 0$ which might be disadvantageous in certain applications. We agree that more information and references on the role of the bias in DNNs and what it means to remove it would strengthen the paper and will include this discussion in the main text.
>
> > **3.** _"Evaluations are not well-motivated. My biggest concern is that several proposed evaluations and their metrics are not well-motivated to show the proposed network architecture is promising."_
>
> We will update the experiments section to better motivate the used evaluations. Essentially, we want the reader to take home the following points: (4.1) Removing the bias is a feasible intervention for some networks that only impairs accuracy marginally. (4.2) Integrated Gradients (and thus $\mathcal{X}$-Gradient) are superior attribution methods compared to regular gradients, and therefore, studying more efficient variants of Integrated Gradients is important. (4.3) Using our method for training with attribution priors, we can compete with the state of the art requiring only a fraction of the computational costs in a concrete example. (4.4) $\mathcal{X}$-DNNs are an interesting class of DNNs that might have advantages in unexpected areas, e.g. for handling contrast changes in vision applications.
>
> > **4.** _"More analysis could be added to justify why Table 3 is important to include and why the result is useful to show the proposed $\mathcal{X}$-DNN is promising."_
>
> That is a good observation and we will make sure that the reasoning becomes more clear. As mentioned in the previous comment the purpose of this experiment is to show that Integrated Gradients (and thus $\mathcal{X}$G) produce higher quality attributions compared to regular gradients and EG(1) [7], and therefore, that studying more efficient variants of Integrated Gradients is important. The finding that Integrated Gradients produces higher quality attributions compared to regular gradients is further acknowledged in reference [2] listed by the reviewer and we will include this reference in the revision.
>
> > **5.** _"Secondly, the paper is motivated to accelerate the training with attribution priors but statistics of the timing and memory cost in training $\mathcal{X}$-DNN are missing in Section 4. More numerical results instead of descriptions, e.g. line 328-330, will show exactly how fast and efficient the proposed algorithm is."_
>
> Memory usage and computation time heavily depend on the used hardware (we use an Nvidia A100 SXM4 (40GB) GPU), and improvement can vary for different experiments. However, we agree that including numerical results is more tangible for the reader and ran an experiment to measure memory usage and computation time of the scenario given in lines 328-330. We train a regular ResNet-50 using Expected Gradients with 32 reference samples and a batch size of 2 and a $\mathcal{X}$-ResNet-50 using $\mathcal{X}$-Gradient and a batch size of 2. The number of reference samples corresponds to the number of reference samples determined in the experiment in Figure 1 (right), where both networks achieve the same accuracy. Results for the GPU memory usage and the required computation time, averaged over 100 iterations, can be found in the table below. For this scenario, we observe a massive improvement by a factor of ~130 for the computation time and ~3.46 for the used GPU memory. We thank the reviewer for the valuable suggestions and will include the experiment in the revision.
>
> | Method    | GPU memory usage (GB) | Computation time (s) |
> |-----------|-----------------------|----------------------|
> | EG [7]   | 14.57                 | 1.1235               |
> | $\mathcal{X}$G (ours) | **4.21**                  | **0.0086**               |
>
> > **6.** _"An example question is: assuming zero vectors as baselines can decourage the network to use features with values close to 0. For example, it will always consider darker pixels as not useful so black dogs may always have smaller attribution values compared to white dogs. Will the $\mathcal{X}$-DNN end up with becoming a network that is always more sensitive to "bright" features and less sensitive to "darker" features that are actually important?"_
>
> We agree that using the $\mathbf{0}$ baseline with Integrated Gradients will yield smaller attribution scores to darker areas which can be a potential issue in certain applications. We thank the reviewer for the observation and will add it as an example of why using the $\mathbf{0}$ baseline might not always be optimal to the revision. However, as this also holds for regular networks we do not believe that $\mathcal{X}$-DNNs by itself will be more sensitive to bright features. Further, one can still use the standard Integrated Gradients method with another baseline for $\mathcal{X}$-DNNs to give more attribution to darker areas. Note that since $F(\mathbf{0}) = 0$ for a $\mathcal{X}$-DNN $F$, the choice of the $\mathbf{0}$ baseline is reasonable in many cases [31].
>
> > **7.** _"Besides, the motivations of using metrics from [1] instead of other work, e.g. [2], are unclear to the reader."_
>
> We have decided to use metrics from [14] instead of metrics from other works because the current state-of-the-art method for attribution priors [7], to which we are comparing, uses the same metrics and we want to ensure consistency. We will add a note to the revision to motivate our reasoning. Further, we will include a reference to reference [2] listed by the reviewer which confirms our findings, as also mentioned in the related comment 4.
>
> We hope that we have been able to adequately clarify the reviewer's concerns and look forward to further discussions.

---

> > ### Comment · Reviewer_gwR5 · 2021-08-26
> > **I appreciate the response and I will increase my score to 7**
> >
> > Dear authors,
> >
> > Thanks for your response and my apology for my late response. I believe most of my questions are resolved and I believe the rest questions in my feedback are good to have but not necessary to include in this paper. I will increase my score from 5 to 7 to reflect this change. I will appreciate If the authors can improve the evaluation sections as we have discussed during the rebuttal session so that the value of your work can be seen more clearly by readers from the community.
> >
> > A minor point to consider if it is worthwhile discussing in the paper based on authors' judgement: there are some other works that discuss the  potential weakness in using zeros as a frequent baseline in Integrated Gradient [1, 2]; therefore, sometimes maybe zero baseline is just simple but might not be a go-to choice as our understandings increase. But I know these work are pretty new so it is not necessary for the authors to be aware of them by the time of this work is submitted to NeurIPS.
> >
> > [1] Wang, Z., Fredrikson, M., & Datta, A. (2021). Boundary Attributions Provide Normal (Vector) Explanations. ArXiv, abs/2103.11257.
> >
> > [2] Pan, D., Li, X., Zhu, D., (2021). Explaining Deep Neural Network Models with Adversarial Gradient Integration. IJCAI 2021

---

### Official Review · Reviewer_bst1 · 2021-07-11

**Rating:** 7
**Confidence:** 3

**Summary:**

The authors propose a class of DNNs, which they call non-negatively homogenous, where the Integrated Gradient (which satisfy certain desirable axioms) for feature attribution can be computed in closed-form using a single gradient evaluation (as opposed to multiple gradients calls for a Riemann sum as in Sundararajan et al 2017). The authors propose instantiating such DNNs in practice by removing the bias term. They conduct experiments on two tasks where they show that removing this bias does not cause a substantial drop in accuracy while still computing IGs efficiently.


**Limitations And Societal Impact:**

Yes, there are two main limitations and the authors acknowledge and address both of them: (i) removing the bias term limits expressivity of the DNN and (ii) the attribution needs a baseline of zero.

**Main Review:**

The paper is well-written and easy to follow. The notation is clear and the propositions/definitions are adequately explained informally which makes the paper easy to understand. Additionally, the authors make an effort to anticipate a reader's questions and answer them which makes it easy to track their specific contributions.

The primary novel contributions are proposing a new class of DNNs that can be instantiated in practice by removing a bias term and proving a closed-form IG for this class. The method is efficient because it only requires a single gradient call.

Re “removing bias term”: The authors sufficiently acknowledge this limitation. It was interesting to see that at least for the experiments in the paper, accuracy does not decline too much. I imagine that in practice, a practitioner could conduct a preliminary analysis to see whether removing the bias hurts performance (through a validation set) and use the proposed DNN if it does not. I think some more suggestions for deciding when this method is appropriate (from a practitioner’s standpoint) would be valuable to add to the paper.

Re “efficiency”: The authors claim that attributions methods need to be invoked at each training step. Does that only happen when attribution priors are used for regularization during training (as in Eq. (1))? Without these priors, it seems to me that attribution only needs to be computed after training a network as opposed to every step. Are there other cases where each training step needs attribution computation?

Table 3 and lines 294-296: The results show that the RAM value for Grad (1) is the best (and does reasonably well for the standard AlexNet too). This is not addressed in the discussion. What are the potential reasons for this?

Line 337 “leads to a strong drop inaccuracy”: I found this surprising given that AlexNet is strictly more expressive than the $\chi$-DNN. I would have imagined that some amount of hyperparameter tuning would have prevented this drop in accuracy.

Figure 2 (right): Are the qualitative attributions shown for the cases where both DNNs predict the correct class label? I am wondering if the less interpretable attribution for AlexNet is simply because it is predicting the wrong label due to the drop in accuracy.

Typo in Eq (5): It should be $i \geq 1$.


**Time Spent Reviewing:**

8

---

> ### Author Response · Authors · 2021-08-10
> **Author response to Reviewer bst1**
>
> We thank the reviewer for the positive and detailed feedback and for valuable suggestions to further improve our work.
>
> > **1.** _"I think some more suggestions for deciding when this method is appropriate (from a practitioner’s standpoint) would be valuable to add to the paper."_
>
> We thank the reviewer for the valuable suggestion and will add the approach proposed by the reviewer to the revision (conducting a preliminary analysis).
>
> > **2.** _"Are there other cases where each training step needs attribution computation?"_
>
> Our method is particularly well suited and developed for including attributions into the training process. Currently, to the best of our knowledge, this is only done for attribution priors. However, attribution priors are still a relatively recent idea that is only just gaining momentum [7,13,20]. In principle, there could be more applications for attributions in the training process that have yet to be discovered.
>
> > **3.** _"Table 3 and lines 294-296: The results show that the RAM value for Grad (1) is the best (and does reasonably well for the standard AlexNet too). This is not addressed in the discussion. What are the potential reasons for this?"_
>
> It is hard to identify an exact reason for this. We can only speculate that since $\mathcal{X}$-Gradient is the same as Grad(1) multiplied with the input, the pixel intensities somehow have a slightly negative impact in this concrete example. Generally, quantifying the quality of an attribution method is not trivial and also depends on the application. This is why we compute and report multiple different metrics to draw a larger picture. We thank the reviewer for the observation and will discuss this issue in the revision.
>
> > **4.** _"Line 337 "leads to a strong drop in accuracy": I found this surprising given that AlexNet is strictly more expressive than the $\mathcal{X}$-DNN. I would have imagined that some amount of hyperparameter tuning would have prevented this drop in accuracy."_
>
> We do not train on images with reduced contrast. Therefore, the networks do not learn how to handle contrast changes (even with extensive hyperparameter tuning) and performance drops for the regular AlexNet. On the other hand, a $\mathcal{X}$-DNN is by design equivariant/invariant to contrast changes, and thus, can handle the reduced contrast inherently without specific data augmentation or retraining.
>
> > **5.** _"Figure 2 (right): Are the qualitative attributions shown for the cases where both DNNs predict the correct class label? I am wondering if the less interpretable attribution for AlexNet is simply because it is predicting the wrong label due to the drop in accuracy."_
>
> For the image with regular contrast, both models correctly predicted 'reflex camera', so the Input x Gradient attribution for the regular AlexNet is less interpretable, even though the class label is predicted correctly. For the low contrast image, the regular AlexNet misclassified the image and we actually display the attributions for the wrong class instead of 'reflex camera'. We will change this and display the attribution for 'reflex camera' for the low contrast image as well. However, the attribution is still less interpretable. We thank the reviewer for pointing out this potential issue.
>
> > **6.** _"Typo in Eq (5): It should be $i \geq 1$."_
>
> We thank the reviewer for pointing this out and will fix it.
>
> We hope that we have been able to adequately clarify the reviewer's concerns and look forward to further discussions.

---

> > ### Comment · Reviewer_bst1 · 2021-08-25
> > **response to authors**
> >
> > Thank you very much for your detailed response. After considering your response and other reviews, I continue to be positive about this paper and would vote for its acceptance.

---

### Official Review · Reviewer_28k9 · 2021-07-18

**Rating:** 4
**Confidence:** 4

**Summary:**

The paper addresses the question of attributing the predictions of deep neural networks to the input features. It proposes to combine a particular restriction on the prediction function: non-negative homogeneity (e.g. a deep neural networks without biases) with the fast gradient x input explanation technique. With this combination, it is possible to produce explanations that satisfy the completeness axiom that can be computed quickly (from a single function evaluation). The paper also argues that non-negative homogeneity makes the model invariant to contrast, which is a useful property in image recognition.

**Limitations And Societal Impact:**

The limitation of methods based on a single gradient evaluation (e.g. for handling functions with multiple scales), as well as the potential cost of having to retrain the model, should be discussed.

**Main Review:**

The paper contributes theoretical results, such as the connection between Integrated Gradients and Gradient x Input for nonnegatively homogeneous functions, the relation between gradient x input and the completeness axiom, as well as the connection between gradient x input and the sensitivity axiom. However, some of these results, in particular, the completeness of gradient x input, are already known and described in earlier papers for the case k=1, which is arguably the most important case since it covers standard deep neural networks. A practical motivation for the higher-order case (k>1) is not provided in the paper.

The paper is mathematically rigorous. However, in Table 1, I'm not sure why Input x Gradient would fail the symmetry and sensitivity(a) tests. I would encourage the authors to provide the corresponding counter-examples in the supplement. Regarding the content of the paper, references need to be added to previous work related to explaining positively homogeneous functions and on the completeness of gradient x input.

The paper is generally clear. However, Table 1 is misleading: Non-negative homogeneity is not an axiom (or some desirable property of the method) but a modeling constraint. Maybe a more suitable line to add in the table would be "computable in a single forward/backward pass". In Table 3, there is no description of what KPM, KNM, etc. mean (only a reference to the paper introducing them). A brief introduction of these metrics would ensure that this table can be fully understood.

The significance of the work is limited. The theoretical contributions are incremental, and the novel aspects (k>1, sensitivity) are not the focus of the experiments. Regarding the practical applicability of the method, removing biases requires retraining the model (which might offset the benefits of reducing gradient evaluations at test time). If the model is instead meant as a self-explainable model, this should be stated explicitly, and the proposed approach should be compared with other self-explainable models, e.g. bag-of-local-features.

Lastly, the proposed X-gradient approach satisfies the completeness axiom, but it is still based on a single gradient evaluation, which raises the question how it could reliably explain a function that contains multiple scales (or some strong local gradient variation superposed to weaker global variations).

**Time Spent Reviewing:**

3

---

> ### Author Response · Authors · 2021-08-10
> **Author response to Reviewer 28k9**
>
> We thank the reviewer for the detailed feedback and valuable suggestions to further improve our work. We would like to clarify the main motivation and contributions of our work under points 1 and 2 below, since some of the comments suggest that we have not made the main points of our work clear enough and as a result, the contributions are considered too small. We apologize for a possible misunderstanding and will ensure that the paper becomes sufficiently self-explaining in the revision.
>
> > **1.** _"The theoretical contributions are incremental, and the novel aspects ($k>1$, sensitivity) are not the focus of the experiments."_
>
> We would like to highlight that our theoretical contributions are not limited to "$k>1$" and "sensitivity". Instead, we find an efficiently computable closed-form solution of Integrated Gradients (a very established attribution method) for a class of DNNs that has a similar capacity as regular DNNs, namely strictly positive homogeneous DNNs of degree k (with $k \geq 1$). We consider a concrete instantiation of this class of DNNs (nonnegatively homogeneous DNNs) that is particularly well suited for training with attribution priors. In two experiments, we demonstrate that this class indeed has almost the same capacity as regular networks while being able to efficiently compute high-quality attributions. This allows us to beat state-of-the-art methods for training with attribution priors while requiring only a fraction of the computing power (Section 4.3). The significance of this contribution is further acknowledged by reviewers gwR5 and bst1.
>
> > **2.** _"Regarding the practical applicability of the method, removing biases requires retraining the model (which might offset the benefits of reducing gradient evaluations at test time)."_
>
> Our method of efficiently computable axiomatic attributions is particularly well suited and developed for scenarios where the attribution is required in the *training* process, such as training with attribution priors. Since including the attribution prior requires retraining the model anyway, retraining due to removing the bias does not add additional overhead. Even though computing the attribution of a regular model at _test_ time, as proposed by the reviewer, is also an important topic, it is not the scope of our work.
>
> > **3.** _"Lastly, the proposed $\mathcal{X}$-Gradient approach satisfies the completeness axiom, but it is still based on a single gradient evaluation, which raises the question how it could reliably explain a function that contains multiple scales (or some strong local gradient variation superposed to weaker global variations)."_
>
> We are unsure what the reviewer means by a function that contains multiple scales and would appreciate some clarification. By Definition 3.5. $\mathcal{X}$-Gradient is only defined for nonnegatively homogeneous DNNs. Per Equation 9 in the supplemental material, those DNNs should not have strong local gradient variations along the straight-line path from the $\mathbf{0}$ baseline to the input $x$.
>
> > **4.** _"The limitation of methods based on a single gradient evaluation (e.g. for handling functions with multiple scales), as well as the potential cost of having to retrain the model, should be discussed."_
>
> Please see comments 2 and 3 above.
>
> > **5.** _"Regarding the content of the paper, references need to be added to previous work related to explaining positively homogeneous functions and on the completeness of gradient x input."_
>
> We included a reference to Ancona et al. [1] who proves the completeness of Input x Gradient for linear models. We will gladly add additional references the reviewer might have in mind to the revision and discuss potential overlaps and differences.
>
> > **6.** _"Non-negative homogeneity is not an axiom (or some desirable property of the method) but a modeling constraint."_
>
> We agree that non-negative homogeneity can be traced back to a modeling constraint for $\mathcal{X}$-Gradient. However, from our perspective, this is not a reason to not consider non-negative homogeneity as an axiom. Indeed, while obviously being axioms, Sensitivity(a) and Completeness, can be traced back to a modeling constraint for $\mathcal{X}$-Gradient as well. Since $\mathcal{X}$-Gradient is only defined for nonnegatively homogeneous DNNs, non-negative homogeneity holds for all $\mathcal{X}$-Gradient attributions which is why we interpret it as a desirable axiom. The reason why it is desirable is related to predictability, as explained in the supplementary material.
>
> > **7.** _"I'm not sure why Input x Gradient would fail the symmetry and sensitivity(a) tests. I would encourage the authors to provide the corresponding counter-examples in the supplement."_
>
> As suggested, we will add a concrete counter-example to why Gradients, and therefore, Input x Gradient violates Sensitivity(a) to the supplemental material similar to the example given in Sundararajan et al. [31]. Regarding symmetry preservation, we accidentally made a wrong entry. Gradients obviously should be symmetry-preserving and we will correct that in the table. We thank the reviewer for pointing this out. Fortunately, this does not affect the rest of the paper.
>
> > **8.** _"Maybe a more suitable line to add in the table would be "computable in a single forward/backward pass"."_
>
> We thank the reviewer for the valuable suggestion and will add the description to the table.
>
> > **9.** _"In Table 3, there is no description of what KPM, KNM, etc. mean (only a reference to the paper introducing them). A brief introduction of these metrics would ensure that this table can be fully understood."_
>
> We thank the reviewer for the valuable suggestion and will make the table more self-explanatory by providing a better description of the metrics.
>
> We hope that we have been able to adequately clarify the reviewer's concerns and look forward to further discussions.

---

### Decision · Program_Chairs · 2021-09-27

**Decision:**

Accept (Poster)

**Comment:**

This paper focuses on feature attribution techniques (which, in a frustrating linguistic turn are often called “explanations”). These techniques are all heuristics chosen for their ability, on the basis of anecdotal evidence to highlight features that the beholder finds important. What if anything they are  “explaining” is seldom explained, and I would encourage the authors to strip the paper of gratuitous uses of the term explanation wherever it is wielded in a quasitechnical form: this includes the title “for Training Neural Networks with Explanations” and in the italicized phrase, as though introducing a technical term: "efficiently explainable”. That said, for better or ill, and the jury is still out, some of these techniques are objects of interest of a large part of the ML community, and while there has been some progress to positing axioms that these attribution maps ought to satisfy, not all methods that satisfy those axioms are known to be efficiently computable.

 In particular, the authors present a new method for computing Integrated Gradients efficiently, showing that when bias terms are removed from neural networks they are efficiently computable. Whether integrated gradients are a useful concept in the first place remains a dubious proposition (what should the baseline be and what basis could anyone possibly have for choosing one?). However, insofar as they are an object of interest and the community is willing to publish lots of papers about them, this paper strikes me as unusually useful and concrete in its offering. The reviewers debated the surprisingness of the finding and whether it was already known to some in the XAI community. However, while I have seen many papers computing integrated gradients, I have no knowledge of anyone computing integrated gradients efficiently via this trick. It seems sufficiently useful, that if it were known, you would have seen it. Questions of the value of the underlying method aside, I do not believe that a finding must require an earthshaking analysis to to be surprising to a great mathematician to be valuable and useful. If that were the case, most useful papers in ML would never have been published. The reviewer’s suggestions that a key result was already known have been rebutted by the authors (who claim that the result is only known for linear models) and the reviewer did not take the opportunity to respond. In this light, I will give the authors the benefit of the doubt.

The authors had a constructive discussion with Reviewers bst1 and gwR5 and I trust that they will act in good faith and add the discussed modifications, clarifications, and exposition to the final draft.  I will recommend acceptance and hope that the authors will act in good faith and tone down the quasitechnical language about “explanation” and to improve the paper per the promises that arose in the discussion period.